# Intra-Articular Facet Joint Injection of Normal Saline for Chronic Low Back Pain: A Systematic Review and Meta-Analysis

**DOI:** 10.3390/medicina59061038

**Published:** 2023-05-28

**Authors:** Areerat Suputtitada, Tanawin Nopsopon, Thanitsara Rittiphairoj, Krit Pongpirul

**Affiliations:** 1Department of Rehabilitation Medicine, Faculty of Medicine, Chulalongkorn University, Rama 4 Road, Pathumwan, Bangkok 10330, Thailand; 2Department of Rehabilitation Medicine, King Chulalongkorn Memorial Hospital, The Thai Red Cross Society, Bangkok 10330, Thailand; doctorkrit@gmail.com; 3Department of Preventive and Social Medicine, Chulalongkorn University, Bangkok 10330, Thailand; tnopsopon@gmail.com; 4Division of Allergy and Clinical Immunology, Brigham and Women’s and Harvard Medical School, Boston, MA 02115, USA; 5Department of Epidemiology, Harvard T.H. Chan School of Public Health, Boston, MA 02115, USA; thanitsara.r@gmail.com; 6Department of International Health, Johns Hopkins Bloomberg School of Public Health, Baltimore, MD 21205, USA; 7Department of Infection Biology & Microbiomes, Faculty of Health and Life Sciences, University of Liverpool, Liverpool L69 3BX, UK

**Keywords:** facet joint injection, chronic low back pain, normal saline, meta-analysis, patient reported clinical outcomes

## Abstract

*Objective*: This systematic review and meta-analysis compared the patient-reported outcomes of intra-articular facet joint injections of normal saline and selected active substances to identify a more effective agent for treating subacute and chronic low back pain (LBP). *Methods*: The PubMed, Embase, Scopus, Web of Science, and CENTRAL databases were searched for randomized controlled trials and observational studies published in English. A research quality assessment was performed using ROB2 and ROBINS-I. A meta-analysis was conducted using a random-effects model, and the mean differences (MD) with 95% confidence intervals (CI) in efficacy outcomes, including pain, numbness, disability, and quality of life, were assessed. Results: Of the 2467 potential studies, 3 were included (247 patients). The active substances and normal saline had similar therapeutic effects on pain within 1 h, after 1–1.5 months, and after 3–6 months, with MD and 95% CI of 2.43 and −11.61 to 16.50, −0.63 and −7.97 to 6.72, and 1.90 and −16.03 to 19.83, respectively, as well as on the quality of life after 1 and 6 months. *Conclusions*: The short- and long-term clinical effects of intra-articular facet joint injections of normal saline are comparable to those of other active substances in patients with LBP.

## 1. Introduction

Low back pain (LBP) is characterized by discomfort, stiffness, or muscular tension between the lower rib edge and buttock creases with or without sciatica (pain radiating from the buttock and downward along the course of the sciatic nerve). Chronic or occasional lower back pain (LBP) is a common musculoskeletal disorder. This is true for people of all ages and countries regardless of whether they are economically developed [1]. In 2019, LBP remained the major cause of years lived with disability (YLDs) worldwide despite a slight decline in the age-standardized prevalence, incidence, and YLDs rate from 1990 to 2019. In 2019, the highest prevalence rates were observed in the 80–84-year-old age bracket for both sexes globally, with the number of cases increasing with age and peaking between the ages of 45 and 54 years [2].

Non-surgical treatment for chronic LBP has been promoted as the first-line treatment, whereas surgical options are considered only when non-surgical treatment is not available or fails. However, more recent studies have shown that spinal fusion is not superior to non-surgical treatment based on the long-term outcomes of pain and disability in patients with chronic LBP [3]. For non-invasive treatments, most guidelines recommend education, exercise, manual therapy, multimodal rehabilitation, and oral medications, including non-steroidal anti-inflammatory drugs and short-term opioids. Intra-articular facet joint injection (FJI), which is considered a minimally invasive procedure, has become common despite the lack of recommendations in recent guidelines [4].

Intra-articular FJI was developed when some authors attempted to identify the pain pattern of facet syndrome by using hypertonic saline and lidocaine as placebos. The injected substance options varied from a commonly used mixture of steroids and local anesthetic agents, steroids alone, and local anesthetic agents alone to more novel substances, including ozone, autologous platelet-rich plasma, and hyaluronic acid [5].

Some recent meta-analyses examined how intra-articular normal saline injections help knee osteoarthritis and found that they reduce pain in the short and long term [6,7]. A network meta-analysis that studied the effectiveness of various substances for intra-articular injection in hip osteoarthritis reported that no active substance was superior to normal saline for pain reduction [8]. Nevertheless, Suputtitada A. recently discovered the use of mechanical needling and sterile water injections to remove calcification and fibrosis. Interestingly, injections of sterile water have a better and longer effect on pain and walking ability than injections of corticosteroids in facet joint syndrome or lidocaine in facet joint syndrome and lumbar spinal stenosis [9,10]. Randomized controlled trials (RCTs) have compared the efficacy of injected substances for intra-articular FJI with normal saline as a placebo control [11,12,13]. This is the first meta-analysis to determine whether the patient-reported outcomes of intra-articular FJIs with normal saline and active substances were comparable. This may change the paradigm for using saline as a placebo.

## 2. Materials and Methods

### 2.1. Study Protocol

This study was conducted according to the recommendations of the Preferred Reporting Items for Systematic Reviews and Meta-analyses (PRISMA) 2020 Statement. This systematic review was registered in the PROSPERO International Prospective Register of Ongoing Systematic Reviews (registration number CRD42020216426).

### 2.2. Search Strategy

PubMed (Appendix A), Embase (Appendix A), CENTRAL (Appendix A), Scopus (Appendix A), and Web of Science (Appendix A) databases were used to search for articles published in English until 1 February 2021. The search strategy is presented in detail in the Appendix A. In addition, the reference lists of the included articles and related citations from other journals were searched for using Google Scholar.

### 2.3. Study Selection

For this systematic review, we worked with an information specialist to design an appropriate search strategy to identify original peer-reviewed RCTs and observational studies evaluating and comparing PROs including pain, numbness, disability, quality of life, and complications associated with FJIs of active substances and normal saline as a placebo in patients diagnosed with chronic LBP. Eligible studies were screened by three independent reviewers (TN, TR, and AS). Discrepancies between the reviewers were resolved by consensus (TN, TR, and AS).

### 2.4. Data Extraction

Data extraction was performed by three independent reviewers (TN, TR, and AS) for the published summary data. Discrepancies between the reviewers were resolved by consensus (TN, TR, and AS).

The following data were extracted: (1) study characteristics (authors, year of publication, study type, journal name, contact information, country, and funding), (2) patient characteristics (sample size, age, age at onset, sex, comorbidities, method of diagnosis, inclusion and exclusion criteria, disease duration, and location of back pain), (3) intervention (type of injected active substances, dosage or regimen of injected substances, type of imaging guide, and co-intervention), (4) comparators (volume of injected normal saline, technique, type of imaging guide, and co-intervention), and (5) outcomes (complete list of the names of all measured outcomes, unit of measurement, follow-up time point, and missing data), as well as any other relevant information. Data were extracted from all relevant text, tables, and figures. We contacted the authors of studies with incomplete data. If the trial authors did not respond within 14 days, analyses were conducted using available data.

### 2.5. Risk of Bias Assessment

The authors worked independently to assess the risk of bias for the included trials using the Cochrane Risk of Bias Tool 2.0 for RCT studies [14]. We assessed the randomization process, deviations from intended interventions, missing outcome data, assessments of outcomes, and selection of the reported results. We graded each domain using the following indicators of bias: low risk, concern, and high risk. For non-randomized trials and observational studies, we used the Risk of Bias in Non-Randomized Studies of Interventions (ROBINS-I) to investigate confounding factors, selection of participants in each study, classification of interventions, deviations from intended interventions, missing data, measurement of outcomes, and selection of the reported results [15]. We rated the domains as follows: low, moderate, serious, and critical risks of bias and no information. As mentioned previously, we contacted the authors if there was insufficient information for the assessment. If the trial authors did not respond within 14 days, assessments were conducted using available data. We resolved this disagreement through discussion.

### 2.6. Statistical Analysis

The primary outcome was the visual analog scale (VAS) score for pain, and the mean difference (MD) between the VAS scores before and after treatment with an associated 95% confidence interval (CI) was determined. Disability outcomes, including the Oswestry Disability Index (ODI) and Pain Disability Questionnaire (PDQ), were also retrieved, including numbness, quality of life outcomes, and adverse events. The results of the studies were included in the meta-analysis and presented as a forest plot, which also showed the statistical power, confidence intervals, and heterogeneity. We assessed clinical and methodological heterogeneity by examining the participant characteristics, intervention regimens, types of interventions, follow-up durations, outcomes, and study design. Statistical heterogeneity was assessed using the I^2^ and χ^2^ statistics. We determined the levels of heterogeneity using the I^2^ statistic, as defined in Chapter 9 of the Cochrane Handbook for Systematic Reviews of Interventions, as follows: 0–40% may not be important, 30–60% may represent moderate heterogeneity, 50–90% may represent substantial heterogeneity, and 75–100% may represent considerable heterogeneity. For missing standard deviation data, we imputed the standard deviation as suggested in Chapter 6 of the Cochrane Handbook for Systematic Reviews of Interventions using what is referred to as the Furukawa methods [16]. Sensitivity analysis was used alternatively to impute the standard deviation, if applicable. A random-effects meta-analysis using the DerSimonian and Laird’s method was performed when clinical, methodological, or statistical heterogeneity was observed. Meta-analysis was performed using Revman 5.3 (Cochrane Collaboration, Oxford, UK).

### 2.7. Patient and Public Involvement

The patients and public were not involved in the design, conduct, reporting, or dissemination of our research.

## 3. Results

### 3.1. Study Selection

A database search identified 2467 records. After removing duplicates, 1305 titles that passed the initial screening, and 112 theme-related abstracts were reviewed to select full-text articles for the eligibility assessment (Figure 1). A total of 109 articles were excluded for the following reasons: wrong study design, 47; wrong comparators, 24; non-peer-reviewed, 23; protocol issues, 6; duplicate, 3; non-English, 3; wrong interventions, 2; and wrong outcomes, 1. Only three studies were eligible based on the inclusion criteria.

### 3.2. Study Characteristics

The three included studies were published between 1989 and 1998 (Table 1) [11,12,13]. All were RCTs. The number of patients per study ranged from 70 to 97, with 247 total (137 were females, 55.5%). The mean age of patients ranged from 43 to 58 years. Two studies reported the following disease durations: a mean of 19.6 months, median of 18 months for the intervention group, and median of 24 months for the normal saline group. None of the included studies documented comorbidities or age at onset. Carette’s study was the only study that used co-intervention for 11 patients receiving corticosteroids and 6 patients receiving placebo [11]. The included patients had chronic LBP for 3–6 months. The location of the back pain also varied from L3/L4 to L5/S1. The follow-up duration ranged from 0.5 h to 6 months.

Three RCTs did not use the same active substances in their respective treatment arms. One study used a mixture of corticosteroids and a local anesthetic as the intervention [12]. One study used corticosteroids alone, and the other used a local anesthetic alone [11,13]. All three trials used fluoroscopy guidance for the FJI.

### 3.3. Quality Assessment

The risk of bias assessment of the three trials in this study showed adequate randomization in one trial, deviations from the intended interventions in two trials, missing outcome data in two trials, adequate measurement of the outcome in three trials, and selection of the reported result in one trial [11,12,13]. A high risk of bias in the selection of the reported result was found for one trial. A summary of the results of the risk of bias assessment of the randomized controlled trials is presented in Figure 2. The percentage of risk of bias in the included randomized controlled trials is presented in Appendix A.

### 3.4. Qualitative Analysis

The primary outcome, assessed using the visual analog scale (VAS), was examined in all the included studies, comprising three randomized controlled trials involving 247 patients. Only the study by Lilius reported a disability outcome using a combined score proposed by the authors for the entire study group; a disability outcome was not reported for each intervention group [12]. Carette’s study reported no difference in overall quality of life represented by the Sickness Impact Profile (SIP) score after the corticosteroid and normal saline FJIs; the score was based on physical and psychosocial characteristics assessed after 1 and 6 months of follow-up, while a significant difference was only reported for the physical dimension after 6 months of follow-up with a favorable outcome of the corticosteroid injection (mean difference (MD) −3.5, 95% CI −6.2 to −0.9) [11]. Two studies reported adverse events [11,12]. The study by Lilius reported seven overall adverse events, with five in men and two in women [13], while Carette’s study reported no major adverse event related to intra-articular FJI [11]. Unfortunately, only one study reported eligible results for each outcome; therefore, a meta-analysis could not be conducted for disability, quality of life, and adverse events. None of the three included trials investigated numbness.

A study by Lilius, et al. demonstrated a significant reduction in subjective pain for all participants at all follow-up points (1 h, 2 weeks, 6 weeks, and 3 months); however, there was no significant difference in the VAS score reduction for intra-articular injection with a mixture of local anesthetics and steroids compared with intra-articular normal saline injection for facet joints L3/4 to L5/S1 for both short- and long-term responses, ranging from 1 h to 3 months after injection [12].

Carette’s study demonstrated no difference in pain reduction after corticosteroid and normal saline injections based on responses to the McGill pain questionnaire [17] after one and six months of follow-up. Self-rated pain assessments by the participants also showed a marked improvement that was not significantly different after 1 month of follow-up; however, patients treated with corticosteroid FJI reported a favorable marked improvement after 6 months of follow-up (MD 31%, 95% CI 14–48%) [11].

Revel, et al. concluded that the FJI of a local anesthetic significantly improved the VAS score for pain relative to the FJI of normal saline (*p* = 0.01) in a specific patient group with five or more of the seven clinical characteristics proposed by the author: (1) age >65 years, (2) no pain exacerbation due to cough, (3) no pain exacerbation by forward flexion, (4) no pain exacerbation when rising from flexion, (5) no pain exacerbation by hyperextension, (6) no pain exacerbation by extension rotation, and (7) pain relief in the recumbent position. For patients who did not satisfy the five criteria, the FJI of normal saline showed better pain improvement, although not significantly [13].

### 3.5. Quantitative Analysis

All three studies that reported pain outcomes assessed by the VAS scale were included in the meta-analysis [11,12,13]. The study by Revel showed a significantly favorable outcome with VAS reduction after FJI of a local anesthetic within 1 h of follow-up [13], whereas the study by Lilius showed a non-significant benefit of normal saline injection [12]. The pooled effect of the two randomized controlled trials showed no difference in pain reduction within 1 h of follow-up after FJIs with active substances and normal saline (MD 2.43, 95% CI −11.61 to 16.50) (Figure 3). The forest plot showing the effect of the active substance versus normal saline on pain reduction within 1 h with an alternative method for standard deviation imputation is presented in Appendix A.

The studies by Lilius and Carette reported no difference in pain after FJIs of active substances and normal saline as a placebo within 1–1.5 months of follow-up [11,12,13]. The meta-analysis of the two studies showed similar results: the mean difference was −0.63 (95% CI −7.97 to 6.72) (Figure 4). The forest plot showing the effect of active substance versus normal saline on pain reduction at 1–1.5 months follow-up visit with an alternative method for standard deviation imputation is presented in Appendix A.

Two studies reported pain scores after long-term follow-up ranging from 3 to 6 months. The study reported significantly greater pain reduction by normal saline than by active substances in the long term [12], while Carette’s study reported significantly greater pain reduction by corticosteroid than by normal saline intra-articular FJI in the long term [11]. The pooled effects of long-term pain reduction after FJIs of active substances and normal saline showed no difference (MD 1.90, 95% CI −16.03 19.83) (Figure 5). The forest plot showing the effect of the active substance versus normal saline on pain reduction at the 3–6 months follow-up visit with an alternative method for standard deviation imputation is presented in Appendix A.

Sensitivity analysis was performed with an alternative method of standard deviation imputation using the standard deviation from a study included in this meta-analysis instead of the previous meta-analysis. The pooled effect of pain reduction within 1 h, after 1–1.5 months of follow-up, and after long-term follow-up was similar to that obtained using the main imputation method.

## 4. Discussion

Our meta-analysis suggests that treatment with intra-articular FJI of normal saline as a placebo showed similar effectiveness as intra-articular FJI of active substances based on patient-reported pain outcomes measured by VAS scales at all studied time points ranging from 0.5 h to 6 months.

It is difficult to draw conclusions from these trials because of the heterogeneity in the age range of the populations, types of active substances, locations of back pain, follow-up times, and outcome measurements. The findings suggested no difference in pain reduction after FJIs with normal saline or active substances in patients with CLBP; however, more robustness is required to provide a high level of evidence.

A study by Lilius, et al. showed significant pain reduction for both FJIs of normal saline and steroids with local anesthetics and no significant differences between intra-articular saline and intra-articular steroids with local anesthetic injections on a subjective pain scale at all follow-up points ranging from 1 h to 3 months. One-fourth of the participants experienced pain reduction for up to three months after the FJI, and the overall disability score significantly improved, regardless of the injected substance. The therapeutic effect of sarin was unexpected, and only suggestions from a psychosocial point of view and self-regression were provided without clear evidence or explanations [12].

Carette’s study demonstrated similar LBP reduction effects of corticosteroid FJI after 1 and 6 months of follow-up, evaluated using the VAS and McGill Pain Questionnaire [17]. The authors concluded that the FJI of corticosteroids provided little benefit to patients with CLBP, considering that normal saline was the true placebo. Moreover, that was the only study included in this meta-analysis that assessed the quality-of-life outcome using the Sickness Impact Profile score; the only favorable effect of steroids was observed for the physical dimension after six months of follow-up, but not for the psychosocial dimension [11].

Revel, et al. explored the characteristics of patients with chronic LBP that were significant predictors of favorable pain reduction after intra-articular FJI with a local anesthetic, and five characteristics of back pain were found. Normal saline was considered a true placebo, and its therapeutic effect was attributable to inadequate diagnostic criteria, which resulted in the false-positive selection of patients with chronic LBP who would potentially benefit from FJI [13].

A high-quality systematic review and meta-analysis was conducted to identify the injection therapy for subacute and chronic LBP, and only one study that compared the FJIs of active substances with placebo was identified and included in a meta-analysis of pain outcomes, with comparisons of short-term and long-term therapeutic effects showing no significant difference [18]. More recent systematic reviews that attempted to compare the efficacies of saline, local anesthetics, and steroids for FJI also identified the same study without conducting a meta-analysis [11]. Thus, our study is the first meta-analysis conducted with more than one included study.

Two meta-analyses that focused on the efficacy of intra-articular normal saline injections for knee osteoarthritis demonstrated therapeutic effects on pain [6,7] and functional outcomes [7]; however, these meta-analyses compared pre- and post-injection effects and not the injected substances. A network meta-analysis that evaluated the effectiveness of various substances for intra-articular injections in patients with hip osteoarthritis showed that intra-articular hip saline injection had similar effects as all other active substances on pain and functional outcome [8]. These studies provided strong evidence for the potential therapeutic effect of intra-articular saline injections, which was consistent with our findings. However, this raises questions regarding the appropriateness of intra-articular injections of normal saline as true placebo.

No early trials validated any significant benefit of the active substance over normal saline as the placebo [11,12,13], which resulted in a lack of supporting evidence for recommending the use of intra-articular FJI in the guidelines. However, the use of intra-articular FJI in the real world has been increasing [5]. The choice of agent for intra-articular FJI is another dilemma. More recent trials have chosen a combination of corticosteroids and local anesthetics or corticosteroids alone as comparators to a novel injected substance, instead of normal saline as a true placebo, despite the lack of evidence of the superior benefit of FJI local anesthetics or corticosteroids over normal saline [5]. Interpretation of the results was difficult, especially when no significant differences were observed. The novel substance had an effect similar to that of corticosteroids, local anesthetics, or their combination as well as normal saline, a true placebo. In the past, several authors have considered normal saline as a true placebo and concluded that there was no therapeutic benefit of intra-articular FJI for LBP [11,13]. The results from our study may be a missing piece of the jigsaw, demonstrating that normal saline was not a true placebo. Thus, intra-articular FJI with normal saline is beneficial for chronic LBP.

Pain reduction by normal saline has been demonstrated in meta-analyses of osteoarthritis involving the knee [6,7] and hip [8]. However, the underlying mechanisms of this effect have been rarely studied and are mostly based on a hypothesis. One hypothesis was that the dilution of inflammatory mediators resulted in pain relief [19]. A study explored other mechanisms including the osmolality effect and sodium concentration, but no sufficient evidence was found to support the hypothesis [20]. For the facet joint, the first saline injection that resulted in pain relief was hypertonic saline [21]. The study by Caterini found that facet joint pain may originate from excessive facet joint fluid [22], which could explain how hypertonic saline, but not normal saline, could relieve facet joint pain. The osmolality effect may be considered because facet joint pain has various causes. For some types of facet joint pain, patients have a normal volume of facet joint fluid but an imbalanced osmolality. This is the only hypothesis, as no available study has explored this question.

Based on these data, it is reasonable to believe that intra-articular FJI of any solution can reduce LBP in patients with spinal stenosis [23,24,25,26,27,28,29]. Due of spurs and cartilaginous metaplasia, successful injection of the facet joint, especially in older adults, may be difficult [9,10]. Owing to the less-than-excellent results of FJI for spinal stenosis, blocking the medial branch of the facet joint has become more common. The outcomes remain to be determined [5,28,29]. Suputtitada A. developed a novel technique that combines mechanical needling with sterile water injection to induce mechanical breakdown of calcification and water jet action to remove calcification and fibrosis at the FJ and surrounding tissues [9,10]. According to this systematic review and meta-analysis, any intra-articular FJI solution may have a water-jet effect, as proposed by Suputtitada A., although it may not be sufficient to eliminate calcification. As a result, the effects of steroids, platelet-rich plasma, hyaluronic, and other solutions were comparable to those of saline, which is interesting. The following are some possible explanations: (1) Every solution was unable to pass through the facet joints due to calcification and fibrosis, causing the effect to be caused solely by needling. (2) Saline has physiological effects on alleviating facet joint pain. (3) The amount of saline may remove calcification and fibrosis, so pain decreases as much as the effect of the solution, which partially passes through the facet joints. (4) The sensitization theory, which still need future investigation.

This meta-analysis has several limitations. First, the heterogeneous characteristics of the patients, including the type of injected substance and timing of outcome assessment, made it difficult to draw conclusions from the data. Second, some trials did not report the standard deviations required for meta-analysis. Thus, standard deviations were imputed, which may not reflect the actual variation in the outcomes of the study. Third, only a few studies focused on injected substances for intra-articular FJI; therefore, only a few studies were included in the meta-analysis, and some PROs had insufficient outcomes to conduct a meta-analysis. Fourth, no new trials have compared the effects of active substances and normal saline administered via FJIs for more than 20 years. This evidence may not be completely relevant to the current practices of intra-articular JFI. However, this study provides the up-to-date evidence to shed light on the therapeutic effects of intra-articular normal saline FJI.

## 5. Conclusions

This systematic review and meta-analysis suggests that the short- and long-term clinical outcomes after intra-articular FJIs with normal saline and other active substances in patients with LBP are comparable. Normal saline may not be considered a placebo. Because desensitization, the eradication of calcification and fibrosis, or spurs make it difficult for substances to reach the facet joint, these are possible reasons why normal saline has an effect that is comparable to that of other substances. This may change the paradigm of using saline as a placebo. Additionally, the efficacy of other substances for regeneration may increase if the calcification and fibrosis have been removed before, which needs further research.

## Figures and Tables

**Figure 1 medicina-59-01038-f001:**
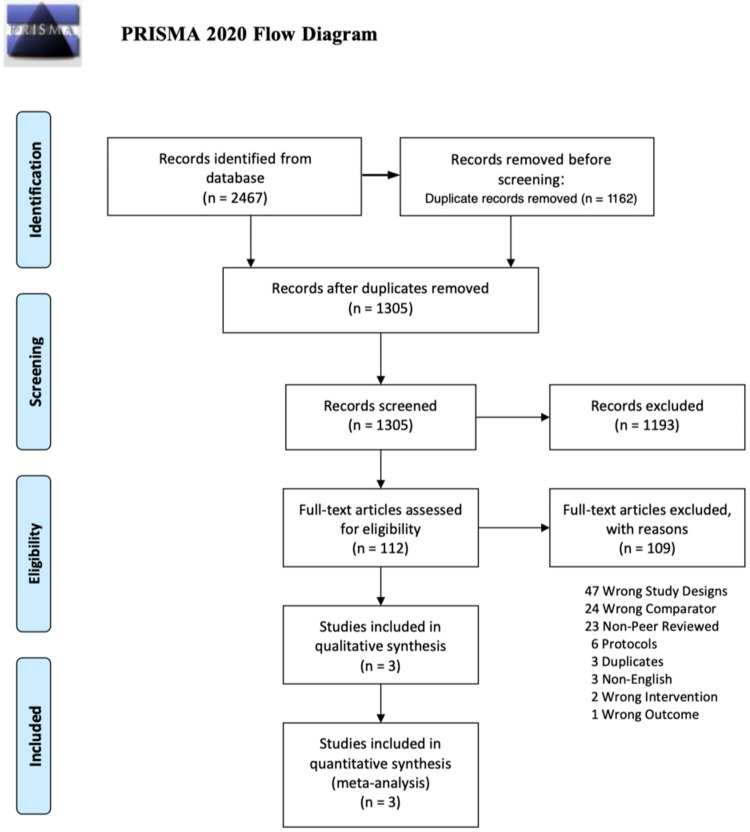
Flow chart diagram for the selection of the studies for analysis based on the Preferred Reporting Items for Systematic Reviews and Meta-analyses (PRISMA) 2020 guidelines.

**Figure 2 medicina-59-01038-f002:**
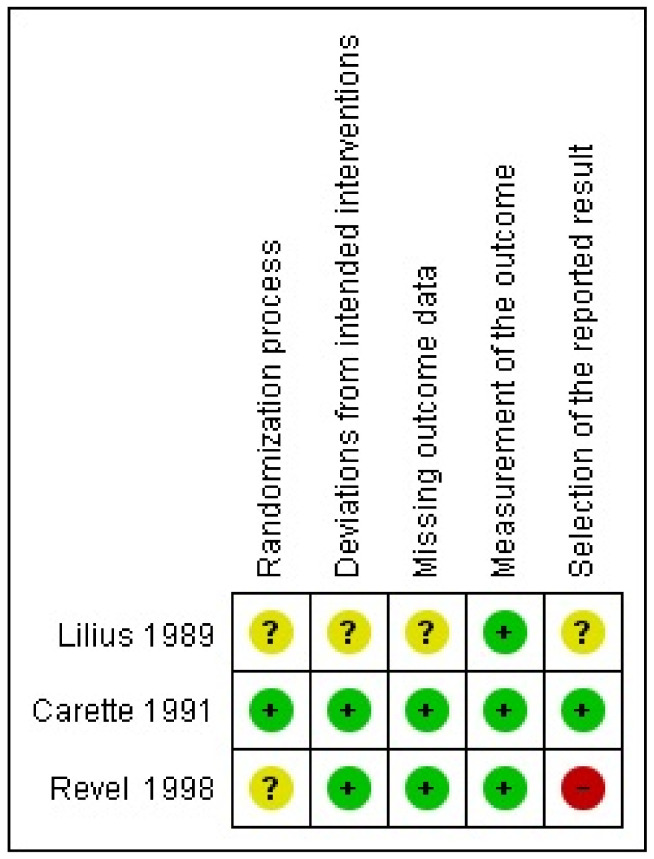
Risks of bias of included randomized controlled trials [11,12,13].

**Figure 3 medicina-59-01038-f003:**
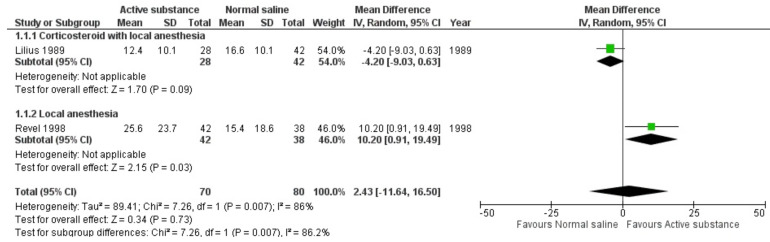
Forest plot showing the effects of active substances versus normal saline on pain within 1 h. [11,12,13]; P = *p*-value.

**Figure 4 medicina-59-01038-f004:**
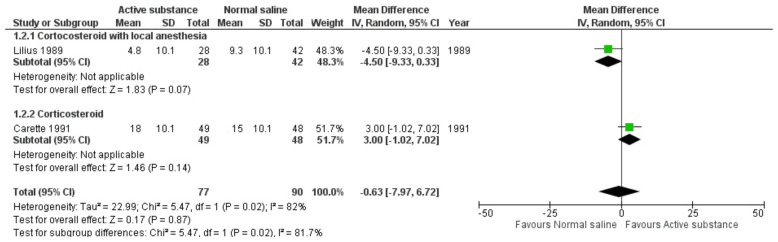
Forest plot showing the effects of active substance versus normal saline on pain after 1–1.5 months of follow-up visits. [11,12,13]; P = *p*-value.

**Figure 5 medicina-59-01038-f005:**
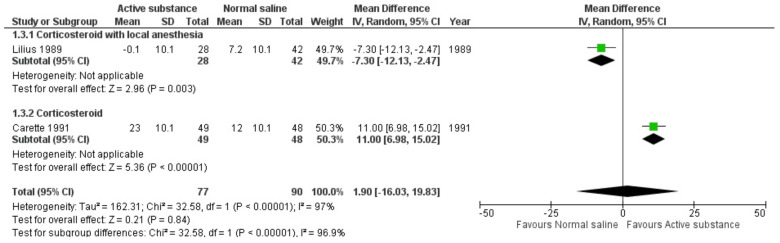
Forest plot showing the effects of active substances versus normal saline on pain after 3–6 months of follow-up visits. [11,12,13]; P = *p*-value.

**Table 1 medicina-59-01038-t001:** Characteristics of the included studies.

	Country	Interventions	Sample Size	Disease Duration (Months)	Female; *n* (%)	Age; Mean (Range)	Imaging Technique	Patient-Reported Outcomes				
								Pain	Disability	Numbness	Quality of Life	Adverse Events
Lilius [12]	Finland	(1) 6 mL bupivacaine hydrochloride mixed with 2 mL methylprednisolone acetate (2) 8 mL physiological saline	70	NR	39 (56%)	44 (19–64)	Fluoroscopy	VAS	Objective disability score (only overall result)	NR	NR	7 overall (5 men, 2 women)
Carette [11]	Canada	(1) 1 mL methylprednisolone acetate mixed with 1 mL of isotonic saline (2) 2 mL isotonic saline	97	(1) median 18 months (2) median 24 months	44 (45%)	43	Fluoroscopy	VAS	NR	NR	SIPs	No adverse events occurred
Revel [13]	France	(1) 1 mL 2% lidocaine (2) 1 mL normal saline	80	Overall mean 19.6 months	54 (68%)	58 (34–87)	Fluoroscopy	VAS	NR	NR	NR	NR

NR: Not reported; VAS: visual analog scale.

## Data Availability

Research data supporting this publication are available from the authors upon reasonable request.

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
