# Peer review of "Intra-Articular Facet Joint Injection of Normal Saline for Chronic Low Back Pain: A Systematic Review and Meta-Analysis"

_medicina, 2023, doi:10.3390/medicina59061038_

Round 1

Reviewer 1 Report

Dear authors, 

congrats for the work!

My comments below:

- Introduction, well written, comprehensive, the mean of the subject is well reflected. My advise: to not use “Paracetamol” but the substance it will be ok.

-Methods: PRISMA  protocol has a newest version, 2020. The version used is from 2009, as it shown in flowchart. Also , there are paper from 1989….from my opinion very old. For such a scientific importance researche to improve clinical practice, maybe the papers analyzed  from the last decade will be more appropriate.

-Results: clear and well written.

-Discussion: there is a risk bias mentioned, discussion are ok.

-Conclusions: can be improve….

Also, there are few refferences for such a comprehensive review. It is mandatory that all the paper mantioned to be in the list.

Author Response

Manuscript medicina-2409429

Intra-Articular Facet Joint Injection of Normal Saline for Chronic Low Back Pain: A Systematic Review and Meta-Analysis

By Areerat Suputtitada * , Tanawin Nopsopon , Thanitsara Rittiphairoj , KRIT Pongpirul

Point-by-point reply to the comments by Reviewer 1

Dear authors, 

congrats for the work!

We are grateful for this comment by the reviewer

My comments below:

- Introduction, well written, comprehensive, the mean of the subject is well reflected.

We are grateful for this comment by the reviewer

My advise: to not use “Paracetamol” but the substance it will be ok.

Line 47-48 We delete “Paracetamol”         including non-steroidal anti-inflammatory drugs,

-Methods: PRISMA  protocol has a newest version, 2020. The version used is from 2009, as it shown in flowchart.

Figure 1 change the content and the legends to PRISMA VERSION 2020  

Also , there are paper from 1989….from my opinion very old. For such a scientific importance researche to improve clinical practice, maybe the papers analyzed  from the last decade will be more appropriate.

OUR Search strategy: PubMed (Appendix 1A), Embase (Appendix 1B), CENTRAL (Appendix 1C) ,Scopus (Appendix 1D, and Web of Science (Appendix 1E) databases were used to search for articles published in English until February 1, 2021. The search strategy is presented in detail in the supplementary material. In addition, the reference lists of the included articles and related citations from other journals were searched for using Google Scholar.  Only three studies in 1989,1991,and 1998 were eligible based on the inclusion criteria.

-Results: clear and well written.

We are grateful for this comment by the reviewer

-Discussion: there is a risk bias mentioned, discussion are ok.

We are grateful for this comment by the reviewer

-Conclusions: can be improve….

This systematic review and meta-analysis suggests that the short- and long-term clinical outcomes after intra-articular FJIs with normal saline and other active substances in patients with LBP are comparable. Normal saline should not be considered a placebo anymore. Because desensitization, the eradication of calcification and fibrosis, or spurs make it difficult for substances to reach the facet joint, these are possible reasons why normal saline has an effect comparable to that of other substances. This will change the paradigm of using saline as a placebo. Importantly, the efficacy of other substances should increase if the calcification and fibrosis have been removed before, which needs further research.

Also, there are few refferences for such a comprehensive review. It is mandatory that all the paper mantioned to be in the list.

All the paper mantioned are in the list.

Thank you very much.

Professor Areerat Suputtitada, MD.

Reviewer 2 Report

This is an excellent meta analysis about intraarticular facet injection with Normal saline.  No additional comments are suggest.

1. What is the main question addressed by the research?

Yes, the main question was addressed by the research, but the main question is less interesting.
2. Do you consider the topic original or relevant in the field? Does it 
address a specific gap in the field?

  No, it is not relevant in the field.
3. What does it add to the subject area compared with other published 
material?  I added only more publications in their study.
4. What specific improvements should the authors consider regarding the 
methodology? What further controls should be considered? Their methodology is very good. No need for improvement.
5. Are the conclusions consistent with the evidence and arguments presented 
and do they address the main question posed?  Yes, the conclusion is consistent with results
6. Are the references appropriate? Yes, it is appropriate.
7. Please include any additional comments on the tables and figures. Table and figures are appropriate.

Quality of English is very good

Author Response

Manuscript medicina-2409429

Intra-Articular Facet Joint Injection of Normal Saline for Chronic Low Back Pain: A Systematic Review and Meta-Analysis

By Areerat Suputtitada * , Tanawin Nopsopon , Thanitsara Rittiphairoj , KRIT Pongpirul

Point-by-point reply to the comments by Reviewer 2

This is an excellent meta analysis about intraarticular facet injection with Normal saline.  No additional comments are suggest.

We are grateful for this comment by the reviewer

  1. What is the main question addressed by the research?

Yes, the main question was addressed by the research, but the main question is less interesting.

This is the first meta-analysis to determine whether the patient-reported outcomes of intra-articular FJIs with normal saline and active substances were comparable.

This is original because the paradigm shift of using normal saline as a placebo is questionable. Importantly, the efficacy of other substances should increase if the calcification and fibrosis have been removed before.

Which add in the conclusion

  1. Do you consider the topic original or relevant in the field? Does it 
    address a specific gap in the field?  No, it is not relevant in the field.

This is original because the paradigm shift of using normal saline as a placebo is questionable. Importantly, the efficacy of other substances should increase if the calcification and fibrosis have been removed before.

Which add in the conclusion

  1. What does it add to the subject area compared with other published 
    material?  I added only more publications in their study.

OUR Search strategy: PubMed (Appendix 1A), Embase (Appendix 1B), CENTRAL (Appendix 1C) ,Scopus (Appendix 1D, and Web of Science (Appendix 1E) databases were used to search for articles published in English until February 1, 2021. The search strategy is presented in detail in the supplementary material. In addition, the reference lists of the included articles and related citations from other journals were searched for using Google Scholar.  Only three studies in 1989,1991,and 1998 were eligible based on the inclusion criteria.

  1. What specific improvements should the authors consider regarding the 
    methodology? What further controls should be considered? Their methodology is very good. No need for improvement.

We are grateful for this comment by the reviewer

  1. Are the conclusions consistent with the evidence and arguments presented 
    and do they address the main question posed?  Yes, the conclusion is consistent with results

We are grateful for this comment by the reviewer

  1. Are the references appropriate? Yes, it is appropriate.

We are grateful for this comment by the reviewer

  1. Please include any additional comments on the tables and figures. Table and figures are appropriate.

We are grateful for this comment by the reviewer

Comments on the Quality of English Language

Quality of English is very good

We are grateful for this comment by the reviewer

Thank you very much.

Professor Areerat Suputtitada, MD.

Round 2

Reviewer 1 Report

The authors made all the changes required.

Author Response

Manuscript medicina-2409429

Intra-Articular Facet Joint Injection of Normal Saline for Chronic Low Back Pain: A Systematic Review and Meta-Analysis

By Areerat Suputtitada * , Tanawin Nopsopon , Thanitsara Rittiphairoj , KRIT Pongpirul

Point-by-point reply to the comments by Reviewer 1, 2nd Round

Comments and Suggestions for Authors

The authors made all the changes required.

We are grateful for this comment by the reviewer

Thank you very much.

Professor Areerat Suputtitada, MD.

Reviewer 2 Report

This is an excellent systemic review and meta analysis on comparable of normal saline and other active substance in aspect of methodology. However, there are only three RCT that complied with their criteria with diversity in their studies as in limitation.

Therefore, authors should not make a conclusion that normal saline is not placebo. But, authors can suggest this hypothesis.

Quality of English is very good.

Author Response

Manuscript medicina-2409429

Intra-Articular Facet Joint Injection of Normal Saline for Chronic Low Back Pain: A Systematic Review and Meta-Analysis

By Areerat Suputtitada * , Tanawin Nopsopon , Thanitsara Rittiphairoj , KRIT Pongpirul

Point-by-point reply to the comments by Reviewer 2 , 2nd Round

Comments and Suggestions for Authors

This is an excellent systemic review and meta analysis on comparable of normal saline and other active substance in aspect of methodology. However, there are only three RCT that complied with their criteria with diversity in their studies as in limitation.

Therefore, authors should not make a conclusion that normal saline is not placebo. But, authors can suggest this hypothesis.

Comments on the Quality of English Language

Quality of English is very good.

We are grateful for this comment by the reviewer

In conclusion

Line 373-374       Change to              Normal saline may not be considered a placebo.

Line 377-379    Change to           This may change the paradigm of using saline as a placebo. Additionally, the efficacy of other substances for regeneration may increase if the calcification and fibrosis have been removed before, which needs further research.

Thank you very much.

Professor Areerat Suputtitada, MD.
